# Comparison of model feature importance statistics to identify covariates that contribute most to model accuracy in prediction of insomnia

**Alexander A. Huang[1], Samuel Y. Huang[2]***

**1** Northwestern University Feinberg School of Medicine, Chicago, IL, United States of America, **2** Virginia Commonwealth University School of Medicine, Richmond, VA, United States of America

* Huangs8@vcu.edu

## Abstract

### Importance

Sleep is critical to a person's physical and mental health and there is a need to create high performing machine learning models and critically understand how models rank covariates.

### Objective

The study aimed to compare how different model metrics rank the importance of various covariates.

### Design, setting, and participants

A cross-sectional cohort study was conducted retrospectively using the National Health and Nutrition Examination Survey (NHANES), which is publicly available.

### Methods

This study employed univariate logistic models to filter out strong, independent covariates associated with sleep disorder outcome, which were then used in machine-learning models, of which, the most optimal was chosen. The machine-learning model was used to rank model covariates based on gain, cover, and frequency to identify risk factors for sleep disorder and feature importance was evaluated using both univariable and multivariable t-statistics. A correlation matrix was created to determine the similarity of the importance of variables ranked by different model metrics.

### Results

The XGBoost model had the highest mean AUROC of 0.865 (SD = 0.010) with Accuracy of 0.762 (SD = 0.019), F1 of 0.875 (SD = 0.766), Sensitivity of 0.768 (SD = 0.023), Specificity of 0.782 (SD = 0.025), Positive Predictive Value of 0.806 (SD = 0.025), and Negative Predictive Value of 0.737 (SD = 0.034). The model metrics from the machine learning of gain and

**Data Availability Statement:** Data Availability: The data from this cohort is freely available without restriction and can be found on the NHANES section of the CDC website. Data Share Statement:

Data described in the manuscript are present at: https://wwwn.cdc.gov/nchs/nhanes/continuousnhanes/default.aspx?cycle=2017-2020.

**Funding:** The author(s) received no specific funding for this work.

**Competing interests:** The authors have declared that no competing interests exist.

cover were strongly positively correlated with one another (r > 0.70). Model metrics from the multivariable model and univariable model were weakly negatively correlated with machine learning model metrics (R between -0.3 and 0).

## Conclusion

The ranking of important variables associated with sleep disorder in this cohort from the machine learning models were not related to those from regression models.

## Introduction

Insomnia is a widespread clinical condition characterized by difficulty initiating or maintaining sleep, which can result in significant physical and mental health consequences. The annual prevalence of insomnia symptoms in the general adult population ranges from 35–50%, with the prevalence of insomnia disorder ranging from 12–20% [1]. Several risk factors contribute to the development of insomnia, including depression, female sex, older age, lower socioeconomic status, concurrent medical and mental disorders, marital status, and race [2–6]. Moreover, insomnia often follows a chronic course, and its functional consequences include reduced productivity, increased absenteeism, and increased healthcare costs. Insomnia also increases the risk of developing mental disorders, including depression, and is associated with worse treatment outcomes in depression and alcohol dependence [3, 7–11]. Furthermore, insomnia is linked to an increased risk of developing metabolic syndrome, hypertension, and coronary heart disease [1, 7, 12–15]. Despite the high prevalence and negative consequences of sleep disorders, researchers are only beginning to apply advanced mathematical models in the field of sleep disorders and it is necessary for physicians to understand how the models are created. Increased research into feature importance will increase the clinical reasoning capacity for physicians.

Linear regression, logistic regression, multivariable statistics, and machine learning have all been essential tools for outcome researchers and physicians in the diagnosis and treatment of various diseases [16, 17]. Linear regression has been used to assess the relationship between continuous predictors and outcomes, which is useful in identifying risk factors for disease progression or treatment outcomes [18]. Logistic regression, on the other hand, has been used to model the probability of binary outcomes, such as the presence or absence of a disease. This technique is particularly useful for diagnosis, where the aim is to correctly classify patients as either having or not having a disease based on various predictors such as symptoms, demographics, and lab values. Multivariable statistics, which include techniques such as multiple regression and analysis of variance, have been used to model the relationship between multiple predictors and an outcome variable [19]. This is important in identifying the most important risk factors for disease progression or treatment outcomes, as well as determining the optimal treatment approach for different patient groups. Machine learning techniques, such as XGBoost and neural networks, have become increasingly popular in recent years due to their ability to handle complex data and identify patterns that may not be apparent with traditional statistical methods [20]. Machine learning has been used to develop predictive models for various diseases, such as diabetes, heart disease, and cancer, as well as to identify subgroups of patients who may benefit from targeted treatments [21–25]. With the rise of machine learning techniques in healthcare research, it is crucial to examine how these models compare to traditional statistical approaches in terms of variable selection and ranking. While traditional

statistical models focus on hypothesis testing and estimation, machine learning models aim to predict outcomes by learning patterns in the data. These differences in approach may lead to variations in variable importance and ranking, which can ultimately impact clinical decision-making. Therefore, to study prediction of insomnia better, it is essential to conduct studies that assess the degree of similarity between machine learning and traditional statistical models in terms of their variable selection and ranking based on various gain statistics, such as cover, frequency, gain, univariable t-statistic, and multivariable t-statistic. This study aims to address this gap in knowledge and provide insight into the similarities and differences between these two modeling approaches.

To address these limitations, we will highlight how some of the most common models researchers use rank various covariates in the model based upon the model statistics. We will utilize a correlation matrix to visually present the correlational relationships between these various model statistics and show the degree of similarity. We will use the NHANES 2017–2020 cohort, a large nationally representative sample of US adults, to analyze demographic, laboratory, physical exam, and lifestyle covariates. This study will help to increase understanding of the different methods for evaluating risk factors for sleep disorders and provide a better understanding of the key risk factors for sleep disorders in the US population. The analysis will utilize the NHANES 2017–2020 cohort, a large, nationally representative sample of US adults, will be used within this study.

## Methods

A cross-sectional cohort study was carried out using the publicly available National Health and Nutrition Examination Survey (NHANES) data. The retrospective study included patients who had completed questionnaires related to their demographics, diet, exercise, and mental health, as well as had undergone laboratory and physical exams. The National Center for Health Statistics (NCHS) Ethics Review Board approved the data acquisition and analysis for this study. To ensure patient privacy, all data including medical records, survey information, and demographic information were fully anonymized prior to analysis. All patients provided their written consent for their data to be made public.

### Dataset and cohort selection

The NHANES program, developed by the NCHS, aims to assess the health and nutritional status of the US population through complex, multi-stage surveys conducted by the CDC. The NHANES dataset includes data on health, nutrition, and physical activity from a representative sample of the US population. For this particular study, the focus was on individuals aged 18 years or older who completed the demographic, dietary, exercise, and mental health questionnaire and had both laboratory and physical exam data available for analysis. All patients in the dataset with full insomnia data were included in this study. 7,929 patients that met the inclusion criteria in this study. A total of 2,302 (29%) of patients had a sleep disorder.

### Assessment of sleep disorder

To identify patients with sleep disorders in this study, we utilized the medical conditions file. Participants were queried with the following question: "Have you ever reported to a healthcare professional or doctor that you experience difficulty sleeping?" If the answer to this question was "Yes," the participant was classified as having a sleep disorder for the purposes of this study.

## Independent variable

The NHANES dataset was searched to identify potential model covariates from the demographics, dietary, physical examination, laboratory, and medical questionnaire datasets. In total, 783 covariates were found and extracted, and they were then merged with the sleep disorder indicator.

## Covariate selection considerations

Recognizing the potential influence of collinearity on variable selection, we underscored the importance of preliminary correlation analysis prior to covariate selection in our revised discussion. To facilitate a more balanced comparison between machine learning and regression models' predictive capabilities for insomnia, we standardized our evaluation approach by incorporating common performance metrics—mean absolute error (MAE), mean squared error (MSE), and mean absolute percentage error (MAPE). This uniformity in performance evaluation enables a more equitable assessment of each model's effectiveness. Furthermore, we enriched our analysis through the inclusion of a residual analysis, examining the discrepancies between observed insomnia probabilities and those predicted by the models. This addition not only enhances the robustness of our findings but also provides deeper insights into the predictive accuracies of the models under consideration, thereby offering a more comprehensive understanding of their utility in the context of insomnia prediction. Through these methodological refinements, our study now presents a nuanced exploration of the comparative advantages of machine learning over traditional regression models in the realm of sleep research, with a particular focus on the selection and ranking of covariates pertinent to insomnia. Given these concerns, we found the most effective methodology was to utilize what has been proposed based upon its simplicity while offering comparable results.

## Model construction and statistical analysis

In this study, univariate logistic models were employed to determine which covariates were associated with a sleep disorder outcome. Covariates that demonstrated a p-value of less than 0.0001 in univariate analysis were included in the final machine-learning model. The use of univariate logistic models served as an initial filter of the 700+ covariates present in the dataset, ensuring that only strong, independent covariates were used in the machine learning models. This initial filtering also facilitated physician review of clinically relevant risk factors. Following the initial filtering process, model importance statistics derived from the machine-learning models were used to identify key risk factors.

Four machine-learning methods were carried out: XGBoost, Random Forest, Adaptive Boost, and Artificial Neural Network. All machine-learning models were constructed using 10-fold cross validation. A train:test (80:20) was used to compute the final set of model fit parameters. The model fit parameters considered in this study included accuracy, F1, sensitivity, specificity, positive predictive value, negative predictive value, and AUROC (Area under the receiver operator characteristic curve).

If it was determined that the models performed differently from one another, the best model based upon the model metrics would be chosen. If the models performed similarly to one another, then the machine learning model of choice would be decided based upon a literature search. In this case, the machine learning model XGBoost was used due to its prevalence within the literature as well as its increased predictive accuracy in healthcare prediction. Furthermore, XGBoost was chosen as the most optimal model based upon the seven model fit parameters that were computed. To identify risk factors for sleep disorder in this study, model covariates were ranked based on three criteria: Gain, Cover, and Frequency. The Gain refers to

the relative contribution of a feature within the machine-learning model, while Cover is the number of observations associated with the feature. Frequency refers to the percentage of times the feature is present in the trees of the model. To visualize the relationship between potential risk factors and sleep disorder, SHAP explanations were utilized. Additionally, feature importance was evaluated using both univariable and multivariable t-statistics.

### Determination of the similarity of the importance of variables by the model metrics

Variables were ranked based on each criterion (Gain, Cover, Frequency, univariable t-statistic, and multivariable t-statistic). A correlation matrix was created that calculated the correlation coefficient between all possible pairings of gain, cover, frequency, univariable t-statistic, and multivariable t-statistic. All statistical analysis was done using R Version 2023.06.0+421 (2023.06.0+421). Packages utilized: dplyr, tidyr, stringr, lubridate, summarytools, psych, ggplot2, plotly, ggpubr, caret, randomForest, glmnet, xgboost, keras, shap, pROC, missForest, boot, cvms, recipes, VennDiagram, fastshap [26].

## Results

### Overall performance and variability of the models

Table 1 shows model accuracy statistics for the four machine learning models. The XGBoost model had strong performance, most notably with the highest mean AUROC of all model metrics with mean AUROC = 0.865 (SD = 0.010), Accuracy = 0.762 (SD = 0.019), F1 = 0.875 (SD = 0.766), Sensitivity = 0.768 (SD = 0.023), Specificity = 0.782 (SD = 0.025), Positive Predictive Value = 0.806 (SD = 0.025), and Negative Predictive Value = 0.737 (SD = 0.034). Among 10,000 simulations completed, we observed that the AUROC ranged from 0.755 to 0.918, a difference of 0.163, the accuracy ranged from 0.657 to 0.894, a 0.237 difference, the F1 ranged from 0.655 to 0.875, a 0.221 difference, the sensitivity ranged from 0.675 to 0.768, a 0.211 difference, and the specificity ranged from 0.565 to 0.936, a 0.370 difference. The machine learning models all had strong performance with mean AUROCs ranging from 0.818 to 0.865.

Table 2 shows the model statistics including the gain, cover, frequency, univariable t-statistic, and multivariable t-statistic for all covariates with p-values <0.0001. These allowed for variable selection and clinical evaluation of the importance of each of these potential features.

Table 3 highlights the top ten variables for each of the model statistics. For all feature importance statistics, PHQ-9 score was the most important with a gain of 0.309, cover of 0.197, frequency of 0.609, univariable t-statistic of 5.536, and multivariable t-statistic of 0.281. Age was the second most important feature importance statistic for 4 out of the 5 a gain of 0.075, cover of 0.094, frequency of 0.061, univariable t-statistic of 5.933, and multivariable t-statistic of 0.177.

### Correlation matrix of correlations between model gain statistics

Fig 1 shows that Gain and Cover were strongly positively correlated with a correlation coefficient of 0.96. Pairs that were moderately positively correlated included gain and frequency with a correlation coefficient of 0.61 as well as cover and frequency with a correlation coefficient of 0.68. The pairs of univariable t-statistic and gain (r = -0.027), univariable t-statistic and cover (-0.067), univariable t-statistic and frequency (r = -0.046), multivariable t-statistic and gain (r = -0.13), multivariable t-statistic and cover (r = -0.16), multivariable t-statistic and frequency (r = -0.23), multivariable t-statistic and univariable t-statistic (r = -0.076) had weakly negative correlations.

**Table 1. Comparison of different machine learning models.**

| XGBoost | Metrics | Minimum | 5th Percentile | 25th Percentile | Median | 75th Percentile | 95th Percentile | Maximum | Mean | Standard Deviation | Range |
|---|---|---|---|---|---|---|---|---|---|---|---|
| | Accuracy | 0.657 | 0.709 | 0.734 | 0.771 | 0.796 | 0.808 | 0.894 | 0.762 | 0.019 | 0.237 |
| | F1 | 0.655 | 0.739 | 0.750 | 0.772 | 0.798 | 0.823 | 0.875 | 0.766 | 0.006 | 0.221 |
| | Sensitivity | 0.675 | 0.727 | 0.770 | 0.774 | 0.791 | 0.847 | 0.887 | 0.768 | 0.023 | 0.211 |
| | Specificity | 0.565 | 0.693 | 0.743 | 0.762 | 0.797 | 0.823 | 0.936 | 0.782 | 0.025 | 0.370 |
| | Positive Predictive Value | 0.668 | 0.731 | 0.761 | 0.785 | 0.819 | 0.863 | 0.941 | 0.806 | 0.025 | 0.273 |
| | Negative Predictive Value | 0.544 | 0.640 | 0.720 | 0.719 | 0.774 | 0.809 | 0.913 | 0.737 | 0.034 | 0.369 |
| | AUROC | 0.755 | 0.800 | 0.833 | 0.840 | 0.861 | 0.905 | 0.918 | 0.865 | 0.010 | 0.163 |
| **Deep Neural Network** | Metrics | Minimum | 5th Percentile | 25th Percentile | Median | 75th Percentile | 95th Percentile | Maximum | Mean | Standard Deviation | Range |
| | Accuracy | 0.645 | 0.715 | 0.736 | 0.744 | 0.789 | 0.804 | 0.879 | 0.766 | 0.020 | 0.234 |
| | F1 | 0.682 | 0.717 | 0.748 | 0.749 | 0.793 | 0.813 | 0.859 | 0.780 | 0.017 | 0.177 |
| | Sensitivity | 0.645 | 0.734 | 0.764 | 0.763 | 0.786 | 0.817 | 0.857 | 0.784 | 0.016 | 0.213 |
| | Specificity | 0.575 | 0.700 | 0.737 | 0.757 | 0.797 | 0.845 | 0.923 | 0.740 | 0.010 | 0.349 |
| | Positive Predictive Value | 0.648 | 0.709 | 0.768 | 0.806 | 0.805 | 0.860 | 0.944 | 0.790 | 0.033 | 0.297 |
| | Negative Predictive Value | 0.532 | 0.657 | 0.711 | 0.703 | 0.732 | 0.806 | 0.896 | 0.722 | 0.027 | 0.365 |
| | AUROC | 0.726 | 0.789 | 0.844 | 0.824 | 0.867 | 0.861 | 0.891 | 0.818 | 0.006 | 0.166 |
| **Random Forest** | Metrics | Minimum | 5th Percentile | 25th Percentile | Median | 75th Percentile | 95th Percentile | Maximum | Mean | Standard Deviation | Range |
| | Accuracy | 0.659 | 0.705 | 0.743 | 0.773 | 0.785 | 0.806 | 0.848 | 0.773 | 0.011 | 0.189 |
| | F1 | 0.671 | 0.733 | 0.736 | 0.762 | 0.782 | 0.816 | 0.869 | 0.770 | 0.017 | 0.198 |
| | Sensitivity | 0.649 | 0.715 | 0.745 | 0.784 | 0.799 | 0.832 | 0.854 | 0.793 | 0.014 | 0.205 |
| | Specificity | 0.559 | 0.669 | 0.750 | 0.762 | 0.765 | 0.816 | 0.902 | 0.753 | 0.018 | 0.344 |
| | Positive Predictive Value | 0.635 | 0.730 | 0.750 | 0.781 | 0.816 | 0.826 | 0.905 | 0.791 | 0.013 | 0.270 |
| | Negative Predictive Value | 0.536 | 0.654 | 0.706 | 0.717 | 0.735 | 0.813 | 0.895 | 0.725 | 0.015 | 0.359 |
| | AUROC | 0.727 | 0.786 | 0.800 | 0.848 | 0.869 | 0.884 | 0.920 | 0.831 | 0.011 | 0.193 |
| **Support Vector Machines** | Metrics | Minimum | 5th Percentile | 25th Percentile | Median | 75th Percentile | 95th Percentile | Maximum | Mean | Standard Deviation | Range |
| | Accuracy | 0.663 | 0.725 | 0.743 | 0.755 | 0.765 | 0.798 | 0.875 | 0.744 | 0.014 | 0.212 |
| | F1 | 0.657 | 0.729 | 0.731 | 0.739 | 0.771 | 0.791 | 0.883 | 0.743 | 0.015 | 0.226 |
| | Sensitivity | 0.640 | 0.719 | 0.773 | 0.780 | 0.776 | 0.828 | 0.889 | 0.796 | 0.017 | 0.249 |
| | Specificity | 0.563 | 0.678 | 0.720 | 0.740 | 0.765 | 0.850 | 0.930 | 0.754 | 0.010 | 0.367 |
| | Positive Predictive Value | 0.665 | 0.722 | 0.769 | 0.791 | 0.840 | 0.832 | 0.929 | 0.800 | 0.020 | 0.264 |
| | Negative Predictive Value | 0.549 | 0.656 | 0.676 | 0.748 | 0.758 | 0.803 | 0.886 | 0.748 | 0.012 | 0.337 |
| | AUROC | 0.724 | 0.812 | 0.820 | 0.857 | 0.848 | 0.858 | 0.904 | 0.839 | 0.014 | 0.180 |

Comparison of four machine learning models (XGBoost, Random Forest, Artificial Neural Network, Adaptive Boosting) using the model statistics: Accuracy, F1, Sensitivity, Specificity, Positive Predictive Value, Negative Predictive Value, and AUROC with the NHANES cohort.

## Discussion

In this retrospective, cross sectional cohort of United States adults, machine learning models utilizing demographic, laboratory, physical examination, and lifestyle questionnaire data all

**Table 2. Model gain statistics.**

| Feature | Gain | Cover | Frequency | Univariable | Multivariable |
|---|---|---|---|---|---|
| PHQ_9 | 0.3088 | 0.1966 | 0.0609 | 5.5356 | 0.2812 |
| Age | 0.0754 | 0.0939 | 0.0607 | 5.9325 | 0.1769 |
| Blood cadmium (ug/L) | 0.0248 | 0.0306 | 0.0408 | 5.4400 | 1.1306 |
| Alcohol..gm. | 0.0253 | 0.0274 | 0.0405 | 0.3661 | 0.4035 |
| BMXWAIST—Waist Circumference (cm) | 0.0270 | 0.0209 | 0.0398 | 11.2737 | 3.4972 |
| BMXWT—Weight (kg) | 0.0299 | 0.0340 | 0.0383 | 0.3272 | 0.3907 |
| Food.folate..mcg. | 0.0245 | 0.0178 | 0.0378 | 6.0746 | 0.9804 |
| RBC folate (ng/mL) | 0.0229 | 0.0213 | 0.0363 | 4.9902 | 1.3090 |
| Caffeine..mg..1 | 0.0234 | 0.0210 | 0.0344 | 5.4546 | 0.4402 |
| Â Red blood cell count (million cells/uL) | 0.0206 | 0.0202 | 0.0341 | 5.6093 | 3.4363 |
| Dietary.fiber..gm. | 0.0196 | 0.0115 | 0.0317 | 5.0440 | 0.1824 |
| HS C-Reactive Protein (mg/L) | 0.0172 | 0.0156 | 0.0310 | 5.7368 | 2.1163 |
| LBXTR. . .Triglyceride..mg.dL. | 0.0186 | 0.0177 | 0.0305 | 12.1990 | 2.0226 |
| Â Glucose, refrigerated serum (mg/dL) | 0.0181 | 0.0146 | 0.0296 | 8.7860 | 1.3297 |
| N-acetyl-S-(n-propyl)-L-cysteine comt | 0.0194 | 0.0197 | 0.0290 | 10.7528 | 1.3323 |
| Red cell distribution width (%) | 0.0165 | 0.0281 | 0.0283 | 3.9850 | 0.6928 |
| Insulin (pmol/L) | 0.0174 | 0.0132 | 0.0277 | 5.7933 | 0.6871 |
| Â Alkaline Phosphatase (ALP) (IU/L) | 0.0149 | 0.0105 | 0.0275 | 5.1057 | 0.2457 |
| Gamma Glutamyl Transferase (GGT) (IU/L) | 0.0141 | 0.0117 | 0.0253 | 5.2053 | 2.2462 |
| BMXBMI—Body Mass Index (kg/m**2) | 0.0155 | 0.0100 | 0.0241 | 5.1035 | 0.6673 |
| Total Protein (g/dL) | 0.0149 | 0.0164 | 0.0237 | 6.7845 | 0.1932 |
| Â Glycohemoglobin (%) | 0.0133 | 0.0154 | 0.0232 | 5.0028 | 0.6921 |
| Blood Urea Nitrogen (mg/dL) | 0.0117 | 0.0090 | 0.0196 | 5.1904 | 1.2308 |
| Cotinine, Serum (ng/mL) | 0.0110 | 0.0094 | 0.0189 | 6.0654 | 1.0683 |
| MCQ366b - Doctor told you to exercise | 0.0386 | 0.0479 | 0.0188 | 6.5157 | 0.0332 |
| Albumin, refrigerated serum (g/dL) | 0.0100 | 0.0076 | 0.0173 | 5.6835 | 2.0091 |
| MCQ540—Ever seen a DR about this pain | 0.0222 | 0.0384 | 0.0171 | 5.0680 | 0.6962 |
| Hydroxycotinine, Serum (ng/mL) | 0.0107 | 0.0096 | 0.0146 | 4.8752 | 1.3299 |
| MCQ300a - Close relative had heart attack? | 0.0083 | 0.0159 | 0.0104 | 2.1012 | 6.3521 |
| MCQ520—Abdominal pain during past 12 months? | 0.0085 | 0.0077 | 0.0086 | 2.4081 | 2.2197 |
| SMQ856—Last 7-d worked at job not at home? | 0.0106 | 0.0196 | 0.0086 | 3.5418 | 0.5547 |
| MCQ300b - Close relative had asthma? | 0.0059 | 0.0183 | 0.0084 | 3.1039 | 1.4597 |
| MCQ160p - Ever told you had COPD, emphysema, ChB | 0.0080 | 0.0165 | 0.0082 | 2.4048 | 2.8528 |
| MCQ366a - Doctor told you to control/lose weight | 0.0097 | 0.0110 | 0.0079 | 1.2951 | 2.2230 |
| MCQ160b - Ever told had congestive heart failure | 0.0063 | 0.0175 | 0.0078 | 0.2023 | 2.8321 |
| MCQ366c - Doctor told you to reduce salt in diet | 0.0070 | 0.0112 | 0.0075 | 2.2783 | 3.4871 |
| MCQ560—Ever had gallbladder surgery? | 0.0060 | 0.0134 | 0.0071 | 1.3450 | 0.9613 |
| SMQ020—Smoked at least 100 cigarettes in life | 0.0040 | 0.0066 | 0.0057 | 0.9800 | 0.5384 |
| Fewer_carbs | 0.0045 | 0.0085 | 0.0055 | 1.6327 | 2.1652 |
| MCQ160m - Ever told you had thyroid problem | 0.0050 | 0.0080 | 0.0054 | 0.0449 | 2.0343 |
| Changed_eating_habits | 0.0040 | 0.0088 | 0.0053 | 14.7371 | 1.0287 |
| MCQ220—Ever told you had cancer or malignancy | 0.0030 | 0.0040 | 0.0043 | 17.9071 | 0.2356 |
| Used_liquid_diet | 0.0027 | 0.0112 | 0.0036 | 11.9125 | 2.6688 |
| MCQ371a - Are you now controlling or losing weight | 0.0021 | 0.0031 | 0.0035 | 13.2250 | 0.3785 |
| MCQ366d - Doctor told you to reduce fat/calories | 0.0022 | 0.0020 | 0.0029 | 6.6864 | 2.8681 |
| Ate_less_junk_food | 0.0019 | 0.0013 | 0.0029 | 4.0939 | 2.8168 |
| MCQ160l - Ever told you had any liver condition | 0.0018 | 0.0072 | 0.0026 | 6.4612 | 7.6485 |

*(Continued)*

**Table 2.** (Continued)

| Feature | Gain | Cover | Frequency | Univariable | Multivariable |
|---|---|---|---|---|---|
| MCQ300c - Close relative had diabetes? | 0.0016 | 0.0011 | 0.0026 | 5.2172 | 2.4245 |
| MCQ160f - Ever told you had a stroke | 0.0015 | 0.0052 | 0.0025 | 7.6107 | 0.4380 |
| MCQ371c - Are you now reducing salt in diet | 0.0012 | 0.0008 | 0.0023 | 11.8760 | 0.1188 |
| Ate_fruits_veg | 0.0012 | 0.0015 | 0.0023 | 0.8459 | 0.0653 |
| MCQ160e - Ever told you had heart attack | 0.0009 | 0.0013 | 0.0016 | 2.7936 | 0.7437 |
| Drank_water_lose_weight | 0.0009 | 0.0004 | 0.0016 | 4.3767 | 1.2387 |
| Gender | 0.0008 | 0.0004 | 0.0015 | 27.6767 | 0.4588 |
| Special_diet_lose_weight | 0.0009 | 0.0014 | 0.0014 | 2.1374 | 0.1151 |
| Supplement_lose_weight | 0.0006 | 0.0025 | 0.0013 | 3.9734 | 1.3689 |
| Ate_less_sugar | 0.0007 | 0.0005 | 0.0012 | 9.6662 | 1.7768 |
| MCQ371d - Are you now reducing fat in diet | 0.0006 | 0.0001 | 0.0012 | 5.0506 | 3.3152 |
| SMQ690A - Used last 5 days—Cigarettes | 0.0005 | 0.0004 | 0.0010 | 11.1669 | 1.6108 |
| MCQ550—Has DR ever said you have gallstones | 0.0002 | 0.0000 | 0.0005 | 5.2151 | 22.8579 |
| MCQ510f - Liver condition: Other liver disease | 0.0002 | 0.0011 | 0.0003 | 4.8092 | 1.9217 |
| MCQ160c - Ever told you had coronary heart disease | 0.0001 | 0.0000 | 0.0003 | 1.2255 | 1.9703 |
| Weight_loss_surgery | 0.0001 | 0.0011 | 0.0002 | 5.5425 | 2.4325 |
| MCQ160d - Ever told you had angina/angina pectoris | 0.0001 | 0.0001 | 0.0002 | 4.1821 | 3.2080 |

The Gain, Cover, and Frequency of all covariates within the XGBoost model. The Gain represents the relative contribution of the feature to the model and is the most important metric of model importance within this study. Covariates ordered according to the Gain statistic.

had strong predictive accuracy with mean AUROCs ranging from 0.818 to 0.865. From the machine learning models the variables with highest associations with a sleep disorder were as follows: depression (PHQ-9), weight, age, and waist circumference. XGBoost was chosen as

**Table 3. Top 10 ranked features for each feature importance method.**

| Feature Importance Method | Gain | Cover | Frequency | Multivariable | Univariable |
|---|---|---|---|---|---|
| **Top 10 Variables Selected** | PHQ_9 | PHQ_9 | PHQ_9 | PHQ_9 | PHQ_9 |
| | Age | Age | Age | Age | MCQ366b - Doctor told you to exercise |
| | MCQ366b - Doctor told you to exercise | MCQ366b - Doctor told you to exercise | Blood cadmium (ug/L) | 'MCQ366b - Doctor told you to exercise' | MCQ366a - Doctor told you to control/lose weight |
| | BMXWT—Weight (kg) | MCQ540—Ever seen a DR about this pain | Alcohol..gm. | 'Albumin, refrigerated serum (g/dL)' | MCQ366d - Doctor told you to reduce fat/calories |
| | BMXWAIST—Waist Circumference (cm) | BMXWT—Weight (kg) | BMXWAIST—Waist Circumference (cm) | 'MCQ520—Abdominal pain during past 12 months?' | BMXBMI—Body Mass Index (kg/m**2) |
| | Alcohol..gm. | Blood cadmium (ug/L) | BMXWT—Weight (kg) | 'BMXWT—Weight (kg)' | MCQ366c - Doctor told you to reduce salt in diet |
| | Blood cadmium (ug/L) | Red cell distribution width (%) | Food.folate..mcg. | GenderMale | MCQ540—Ever seen a DR about this pain |
| | Food.folate..mcg. | Alcohol..gm. | RBC folate (ng/mL) | Weight_loss_surgery | Age |
| | Caffeine..mg..1 | RBC folate (ng/mL) | Caffeine..mg..1 | 'SMQ856—Last 7-d worked at job not at home?' | SMQ856—Last 7-d worked at job not at home? |
| | RBC folate (ng/mL) | Caffeine..mg..1 | Â Red blood cell count (million cells/uL) | 'MCQ371c - Are you now reducing salt in diet' | BMXWT—Weight (kg) |

**Description:** SHAP explanations, covariate value on the x-axis, change in log-odds on the y-axis, red line represents the relationship between the covariate and log-odds for insomnia. Splines, the relationship between the covariate value on the x-axis and the probability for insomnia on the y-axis.

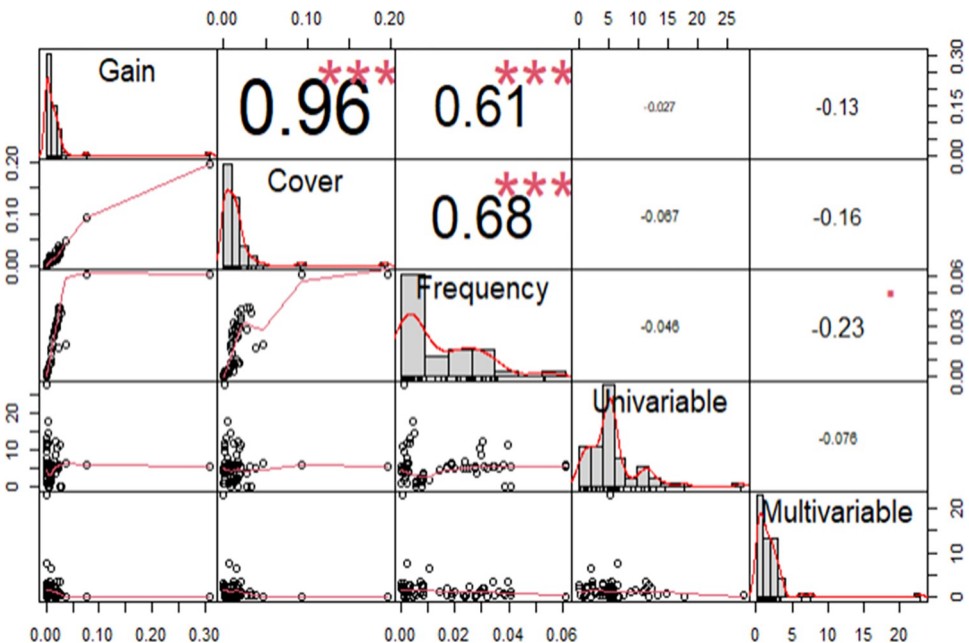

**Fig 1. Comparisons of model gain statistics.**

the machine learning model of choice because it had the highest mean AUROC. It was important to compare various machine learning models to show that the performance metrics of the different machine learning models are similar across the distribution to ensure that any differences in variable contributions are not due to differences in model performance.

In the field of machine learning, identifying the most important variables in predicting an outcome is crucial. Our study reveals that different measures of feature importance result in wide variability in selecting the top 10 covariates in the final model. This discrepancy arises from the varied methods used to assess which covariates contribute the most to the model, such as linear regression's reliance on a least squares metric that considers estimates as non-interactive. In contrast, machine learning models construct feature importance metrics using gain cover and frequency statistics, resulting in a different set of top covariates [17]. The interaction of these biomolecular pathways is challenging to comprehend, and traditional regression models may not effectively account for these complex interactions [27]. Therefore, we propose that machine learning methods utilizing gain cover and frequency model selection statistics are better equipped to handle these complexities and provide a more accurate representation of the most important covariates in predicting outcomes.

In the context of feature selection in machine learning, it is crucial to recognize that each method yields a different set of best covariates. As such, different model selection statistics need to be combined to determine the best approach. In this study, we evaluated three measures for machine learning feature importance, including cover, gain, and frequency, as well as two measures for regression, including univariable T statistics and multivariable T statistics. We found a strong correlation between the different machine learning models of frequency, gain, and cover. However, there were weak and sometimes negative correlations between the feature importance ranks of machine learning models and those of univariable and multivariable regression. These findings suggest that there are complex interactions happening within the machine learning models that are not accounted for in the multivariable regression. As

such, interpreting the multivariable results may yield an inaccurate representation of the importance of these covariates [28].

In modeling, understanding which covariates are important due to their interactions with other covariates or on their own is challenging. Accounting for confounding variables has always been difficult, and multivariable regression is the most common approach, but it cannot efficiently account for every possible interaction. It is impossible to run all the pairwise, three-way, four-way, and five-way interactions present in a multivariable model efficiently [29]. Thus, the most efficient way to capture these interactions is through machine learning models that iterate through the data, develop the most efficient models, and are cross-validated and effectively tested through train-test splits. Therefore, the large discrepancy between feature ranks between univariable and multivariable regression and that of machine learning models, highlights the importance of accounting for interaction terms through machine learning methods.

Our study evaluates the differences between interaction terms and how they lead to differences in model feature statistic rankings. By efficiently visualizing the relationship between each covariate and accounting for all confounding variables through machine learning, we can better investigate and identify the most important variables for further investigation in prospective studies. Therefore, we argue that machine learning brings a new way of evaluating variables beyond traditional regression, as it can account for confounding variables and identify important variables for future studies.

## Limitations

This study has both strengths and limitations. The utilization of the NHANES dataset, which is a large retrospective cohort, allows for the selection of a substantial sample size, evaluation of data quality, and broad generalizability. However, it also carries the limitations of retrospective studies, such as reliance on self-reported surveys to obtain information on the outcome of interest and lifestyle choices. Prospective studies with automated measurements of foods may be more accurate, but they may not have the advantage of including a larger volume of participants through self-reported information. Another limitation is the voluntary nature of the cohort, which may introduce selection bias. However, the demographic diversity of the cohort analyzed suggests that the findings may still be generalizable to other cohorts. It is important to note that while this study focused on machine learning models and traditional statistical models, other models that are not linear or involve machine learning could be explored in future studies.

## Conclusion

Machine learning models offer additional information in ranking variable importance for predicting insomnia in addition to regression models.

## Author Contributions

**Conceptualization:** Samuel Y. Huang.

**Formal analysis:** Alexander A. Huang.

**Investigation:** Alexander A. Huang, Samuel Y. Huang.

**Methodology:** Samuel Y. Huang.

**Resources:** Alexander A. Huang.

**Software:** Alexander A. Huang.

**Supervision:** Alexander A. Huang.

**Validation:** Alexander A. Huang.

**Visualization:** Alexander A. Huang.

**Writing – original draft:** Samuel Y. Huang.

**Writing – review & editing:** Samuel Y. Huang.

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
