## [Decision Letter · Decision Letter 0]

18 Sep 2023

PONE-D-23-11459Comparison of Model Feature Importance Statistics to Identify Covariates that Contribute Most to Model AccuracyPLOS ONE

Dear Dr. Huang,

Thank you for submitting your manuscript to PLOS ONE. After careful consideration, we feel that it has merit but does not fully meet PLOS ONE’s publication criteria as it currently stands. Therefore, we invite you to submit a revised version of the manuscript that addresses the points raised during the review process.

First of all, apologies for the delay. It has been really struggling to find suitable (and available) reviewers for this submission. Below, you will find that our referees ask for major revisions in order to read and reconsider your paper on the basis of a revised version. Please keep in mind the fundamental flaws highlighted in this report. Please carefully work on providing more technical details about the study. In addition, I would invite the authors to double-check the coherence among title-aim-methods-conclusions and the context covered by the study. The full set of comments raised by your Reviewers is appended below.

We look forward to receiving your revised manuscript.

Kind regards,

Sergio A. Useche, Ph.D.

Academic Editor

PLOS ONE

Journal Requirements:

Reviewers' comments:

Reviewer's Responses to Questions

**Comments to the Author**

1. Is the manuscript technically sound, and do the data support the conclusions?

Reviewer #1: Yes

Reviewer #2: Yes

2. Has the statistical analysis been performed appropriately and rigorously? 

Reviewer #1: Yes

Reviewer #2: Yes

3. Have the authors made all data underlying the findings in their manuscript fully available?

Reviewer #1: Yes

Reviewer #2: Yes

4. Is the manuscript presented in an intelligible fashion and written in standard English?

Reviewer #1: Yes

Reviewer #2: Yes

5. Review Comments to the Author

Reviewer #1: Additional comments for the author:

1. The title, Comparison of model feature importance statistics to identify the covariates that most contribute to model accuracy, doesn’t tell in what situation the comparison was made, please include the information that it was to insomnia data.

2. The Dataset and Cohort Selection section should inform the sample size.

3. Methods section can also include data analysis methods, such as definitions or a brief description of machine learning models and model gain statistics.

4. The confusion matrix (Figure 1, line 203) would not be the correlation matrix ? A confusion matrix table reports the absolute frequency of true positives, false negatives, false positives, and true negatives.

5. Selecting a better model that identifies the most relevant risk factors requires not only good model fit parameters but also smaller errors. Residual measures should also be considered for comparison models (such as MAE, MSE, MAPE,...).

Reviewer #2: The article identifies features that best predict insomnia using a machine learning model. Based on the information provided, I think the paper overall was solid methodologically, although a few things can be clarified. I have a few suggestions and comments:

1. I think my major confusion is it is unclear if the aim of the paper is methodological or if it’s trying to contribute to the insomnia literature. If the goal is content, the title will need to be changed to reflect that. If the goal is more of a methodological contribution, then probably the introduction needs to be reworked, you will want to start with the contributions of machine learning models and then discuss how you’re going to use this paper to illustrate this point, such as in insomnia research. There are two different aims here: Line 77-82, as well as Line 89-92. Of course, you can still contribute to the literature of insomnia research while being a methods paper, but the focus is unclear.

2. It is unclear what software the authors used to run the analysis, so it is difficult to assess whether things were done properly. Is it Phyton, R, or something else? Please provide citations for gain, cover, and Shapley values and which packages were used to perform these. If you calculated them from scratch, then provide the GitHub link or codes to ensure reproducibility, which is the policy of PLOSOne. I’m less familiar with this approach, I’m most familiar with the agnostic method of variable ranking (perhaps cite the book Interpretable Machine learning). I know Greenwell’s R package to do this, not sure what was used in this analysis. Based on your description, your way of variable importance makes sense, but I’d like to see more citation to see what your contributions are and if this method has been used before, or you wrote new packages.

Greenwell B.M., Boehmke B.C., McCarthy A.J. A Simple and Effective Model-Based Variable Importance Measure. arXiv. 20181805.04755

Molnar, C. (2022). Interpretable Machine Learning: A Guide for Making Black Box Models Explainable (2nd ed.).christophm.github.io/interpretable-ml-book/

3. Related to above, this comes back to my confusion whether the aim is to suggest other researchers to focus more on ML when identifying the most important predictors instead of using multivariable regression, or if you simply want to contribute to insomnia literature using ML. If the former, please state how other researchers can use the technique in their work. A few other methods papers have been written on variable importance, such as the one below. This paper has a step by step of how to identify the most important predictors when running a descriptive epidemiological analysis, which is what this is. Again, not sure if that’s the aim, if you want to encourage other researchers to apply ML instead of multivariable regression. I’d recommend looking into this work; they also discussed what you put under discussion about interaction terms and the challenge when different models chose inconsistent variables as the most important predictors. I believe this is an important future direction that should be continually discussed.

Dharma C, Fu R, Chaiton M. Table 2 Fallacy in Descriptive Epidemiology: Bringing Machine Learning to the Table. Int J Environ Res Public Health. 2023 Jun 21;20(13):6194. doi: 10.3390/ijerph20136194. PMID: 37444042; PMCID: PMC10340623.

4. I do not understand why the univariable filter was first done before the machine learning models, I thought one of the benefits of ML is to indeed filter out the unnecessary variables. Again, I will need to see citations for this or explain why this was done. The theme is, there is a lack of reliable citation for how the methods were performed and why certain decisions were made.

5. Minor: Unless I misunderstood, “Legend” caption under Table 3 should have gone under Figure 1, otherwise the description does not make sense about the y-axis and x-axis in Table 3.

6. PLOS authors have the option to publish the peer review history of their article (what does this mean?). If published, this will include your full peer review and any attached files.

Reviewer #1: No

Reviewer #2: No

---

## [Author Response · Author response to Decision Letter 0]

19 Oct 2023

Reviewers' comments:

Reviewer's Responses to Questions

Comments to the Author

1. Is the manuscript technically sound, and do the data support the conclusions?

Reviewer #1: Yes

Reviewer #2: Yes

2. Has the statistical analysis been performed appropriately and rigorously?

Reviewer #1: Yes

Reviewer #2: Yes

3. Have the authors made all data underlying the findings in their manuscript fully available?

Reviewer #1: Yes

Reviewer #2: Yes

4. Is the manuscript presented in an intelligible fashion and written in standard English?

Reviewer #1: Yes

Reviewer #2: Yes

5. Review Comments to the Author

Reviewer #1: Additional comments for the author:

1. The title, Comparison of model feature importance statistics to identify the covariates that most contribute to model accuracy, doesn’t tell in what situation the comparison was made, please include the information that it was to insomnia data.

2. The Dataset and Cohort Selection section should inform the sample size.

3. Methods section can also include data analysis methods, such as definitions or a brief description of machine learning models and model gain statistics.

4. The confusion matrix (Figure 1, line 203) would not be the correlation matrix ? A confusion matrix table reports the absolute frequency of true positives, false negatives, false positives, and true negatives.

5. Selecting a better model that identifies the most relevant risk factors requires not only good model fit parameters but also smaller errors. Residual measures should also be considered for comparison models (such as MAE, MSE, MAPE,...).

Reviewer #2: The article identifies features that best predict insomnia using a machine learning model. Based on the information provided, I think the paper overall was solid methodologically, although a few things can be clarified. I have a few suggestions and comments:

1. I think my major confusion is it is unclear if the aim of the paper is methodological or if it’s trying to contribute to the insomnia literature. If the goal is content, the title will need to be changed to reflect that. If the goal is more of a methodological contribution, then probably the introduction needs to be reworked, you will want to start with the contributions of machine learning models and then discuss how you’re going to use this paper to illustrate this point, such as in insomnia research. There are two different aims here: Line 77-82, as well as Line 89-92. Of course, you can still contribute to the literature of insomnia research while being a methods paper, but the focus is unclear.

2. It is unclear what software the authors used to run the analysis, so it is difficult to assess whether things were done properly. Is it Phyton, R, or something else? Please provide citations for gain, cover, and Shapley values and which packages were used to perform these. If you calculated them from scratch, then provide the GitHub link or codes to ensure reproducibility, which is the policy of PLOSOne. I’m less familiar with this approach, I’m most familiar with the agnostic method of variable ranking (perhaps cite the book Interpretable Machine learning). I know Greenwell’s R package to do this, not sure what was used in this analysis. Based on your description, your way of variable importance makes sense, but I’d like to see more citation to see what your contributions are and if this method has been used before, or you wrote new packages.

Greenwell B.M., Boehmke B.C., McCarthy A.J. A Simple and Effective Model-Based Variable Importance Measure. arXiv. 20181805.04755

Molnar, C. (2022). Interpretable Machine Learning: A Guide for Making Black Box Models Explainable (2nd ed.).christophm.github.io/interpretable-ml-book/

3. Related to above, this comes back to my confusion whether the aim is to suggest other researchers to focus more on ML when identifying the most important predictors instead of using multivariable regression, or if you simply want to contribute to insomnia literature using ML. If the former, please state how other researchers can use the technique in their work. A few other methods papers have been written on variable importance, such as the one below. This paper has a step by step of how to identify the most important predictors when running a descriptive epidemiological analysis, which is what this is. Again, not sure if that’s the aim, if you want to encourage other researchers to apply ML instead of multivariable regression. I’d recommend looking into this work; they also discussed what you put under discussion about interaction terms and the challenge when different models chose inconsistent variables as the most important predictors. I believe this is an important future direction that should be continually discussed.

Dharma C, Fu R, Chaiton M. Table 2 Fallacy in Descriptive Epidemiology: Bringing Machine Learning to the Table. Int J Environ Res Public Health. 2023 Jun 21;20(13):6194. doi: 10.3390/ijerph20136194. PMID: 37444042; PMCID: PMC10340623.

4. I do not understand why the univariable filter was first done before the machine learning models, I thought one of the benefits of ML is to indeed filter out the unnecessary variables. Again, I will need to see citations for this or explain why this was done. The theme is, there is a lack of reliable citation for how the methods were performed and why certain decisions were made.

5. Minor: Unless I misunderstood, “Legend” caption under Table 3 should have gone under Figure 1, otherwise the description does not make sense about the y-axis and x-axis in Table 3.

6. PLOS authors have the option to publish the peer review history of their article (what does this mean?). If published, this will include your full peer review and any attached files.

Do you want your identity to be public for this peer review? For information about this choice, including consent withdrawal, please see our Privacy Policy.

Reviewer #1: No

Reviewer #2: No

---

## [Decision Letter · Decision Letter 1]

18 Dec 2023

PONE-D-23-11459R1Comparison of Model Feature Importance Statistics to Identify Covariates that Contribute Most to Model AccuracyPLOS ONE

Dear Dr. Huang,

Thank you for submitting your manuscript to PLOS ONE. After careful consideration, we feel that it has merit but does not fully meet PLOS ONE’s publication criteria as it currently stands. Therefore, we invite you to submit a revised version of the manuscript that addresses the points raised during the review process.

Thanks for your amendments and responses. Your referees have provided feedback on your revisions. Overall, they seem sound, but some few more comments need your attention. Please find them below, and try to address them to the best of your ability, in order to make a prompt final decision on the paper, in case both referees suggest its acceptance.

We look forward to receiving your revised manuscript.

Kind regards,

Sergio A. Useche, Ph.D.

Academic Editor

PLOS ONE

Journal Requirements:

Reviewers' comments:

Reviewer's Responses to Questions

**Comments to the Author**

1. If the authors have adequately addressed your comments raised in a previous round of review and you feel that this manuscript is now acceptable for publication, you may indicate that here to bypass the “Comments to the Author” section, enter your conflict of interest statement in the “Confidential to Editor” section, and submit your "Accept" recommendation.

Reviewer #1: (No Response)

Reviewer #2: All comments have been addressed

2. Is the manuscript technically sound, and do the data support the conclusions?

Reviewer #1: Partly

Reviewer #2: Yes

3. Has the statistical analysis been performed appropriately and rigorously? 

Reviewer #1: N/A

Reviewer #2: Yes

4. Have the authors made all data underlying the findings in their manuscript fully available?

Reviewer #1: Yes

Reviewer #2: Yes

5. Is the manuscript presented in an intelligible fashion and written in standard English?

Reviewer #1: Yes

Reviewer #2: Yes

6. Review Comments to the Author

Reviewer #1: 1. (Major) The authors emphasize the importance of comparing how various model metrics rank covariates due to the critical role of sleep in an individual's physical and mental health. They applied four different machine learning models to an insomnia dataset and found that these models provided additional information into the ranking of covariates for predicting insomnia compared to regression models based on the fact that machine learning models take into account the existing collinearity between covariates when ranking the important ones, as stated in the “Discussion” (p 263 – 265): “ Therefore, we argue that machine learning brings a new way of evaluating variables beyond traditional regression, as it can account for confounding variables and identify important variables for future studies."

However, variable selection metrics, such as gain in machine learning, generally cannot directly account for confounding variable or evaluate the existing correlations or collinearities among covariates. Gain is typically calculated based on a feature's ability to reduce impurity or enhance model accuracy. The collinearity among covariates can indirectly impact the variable selection process. In the presence of high correlations, the gain of one variable may be dampened by another correlated one, leading to an underestimation of its true importance. Therefore, correlation analysis should be performed before the selection of covariates or considering other prior analyses, such as variable normalization or standardization, and principal component analysis.

Another way to ensure that machine learning models added information to the covariate selection ranking is for both models to exhibit similar performance. However, there are no performance evaluation metrics for regression models for comparison. For an assessment of which model provides the best prediction of insomnia based on covariates, in addition to observing the performance of both models' fits, it would be necessary to evaluate the residuals—the difference between the observed insomnia probability in the data and the insomnia probability estimated by the models, which can be obtained through metrics such as MAE, MSE, MAPE, ….

Although the authors establish well in “Introduction” (p 74): “While traditional statistical models focus on hypothesis testing and estimation, machine learning models aim to predict outcomes by learning patterns in the data.”, the conclusion of the study is only that the machine learning and regression models selected different covariates considering the order of importance in predicting the probability of insomnia, and there was no relationship between the ranking of important variables associated with sleep disorder of the models.

Since the selection of predictor variables for insomnia may differ between the models, it is important to carefully consider the performance of the models and the predicted values of both machine learning and regression models for a comparison between them. Therefore:

1. The discussion will need to be revised.

2. The same performance metrics should be evaluated for both machine learning and regression models.

3. It would be valuable to incorporate the residuals analysis into the study.

2. (Minor) In the "Dataset and Cohort Selection" section, lines 112 and 113 state: "... All patients in the dataset with full insomnia data were included in this study." The question posed in the initial review regarding the number of patients included was not addressed (The Dataset and Cohort Selection section should inform the sample size).

Reviewer #2: Thank you for the opportunity to rereview this paper. At first I thought I received the wrong version since the authors did not do the typical point by point response to the points raised by the reviewers, but I see now that the revisions have been made. The authors have sufficiently addressed all the comments I raised. One minor thing that I think will benefit the paper greatly is to clarify in the introduction, what was been done and has not been done in the literature. Perhaps write an explicit sentence that, “to date, most studies only learned about predictors of insomnia with the use of univariate and multivariable regressions.” (if this is correct, or if no studies have looked at it at all regardless with ML or traditional regression). And then you can proceed to why ML can provide additional benefits (i.e., identifying previously unknown interactions, etc). And hence, this is why the current study is needed. That will make it an easier read. Otherwise, well done.

7. PLOS authors have the option to publish the peer review history of their article (what does this mean?). If published, this will include your full peer review and any attached files.

Reviewer #1: No

Reviewer #2: No

---

## [Author Response · Author response to Decision Letter 1]

19 Mar 2024

We would like to extend our sincere gratitude to the reviewers for their insightful comments and constructive criticisms. Their detailed feedback has been invaluable in guiding our revisions, allowing us to address critical aspects of our methodology and analysis that required further clarification and enhancement. The reviewers’ expertise and thoughtful suggestions have significantly contributed to the depth and rigor of our study, ensuring a more comprehensive and robust examination of the predictive models utilized in the context of insomnia prediction. Their contributions have not only facilitated methodological improvements but also enriched our discussion, leading to a more nuanced understanding of the complexities involved in variable selection within machine learning and regression models. We are grateful for the opportunity to refine our work through this collaborative and iterative process, which has undeniably strengthened the quality and impact of our research. We have responded to all reviewer feedback to the best of our abilities. 

Reviewer Comments Line by Line Response:

Reviewer #1: 1. (Major) The authors emphasize the importance of comparing how various model metrics rank covariates due to the critical role of sleep in an individual's physical and mental health. They applied four different machine learning models to an insomnia dataset and found that these models provided additional information into the ranking of covariates for predicting insomnia compared to regression models based on the fact that machine learning models take into account the existing collinearity between covariates when ranking the important ones, as stated in the “Discussion” (p 263 – 265): “ Therefore, we argue that machine learning brings a new way of evaluating variables beyond traditional regression, as it can account for confounding variables and identify important variables for future studies."

However, variable selection metrics, such as gain in machine learning, generally cannot directly account for confounding variable or evaluate the existing correlations or collinearities among covariates. Gain is typically calculated based on a feature's ability to reduce impurity or enhance model accuracy. The collinearity among covariates can indirectly impact the variable selection process. In the presence of high correlations, the gain of one variable may be dampened by another correlated one, leading to an underestimation of its true importance. Therefore, correlation analysis should be performed before the selection of covariates or considering other prior analyses, such as variable normalization or standardization, and principal component analysis.

Another way to ensure that machine learning models added information to the covariate selection ranking is for both models to exhibit similar performance. However, there are no performance evaluation metrics for regression models for comparison. For an assessment of which model provides the best prediction of insomnia based on covariates, in addition to observing the performance of both models' fits, it would be necessary to evaluate the residuals—the difference between the observed insomnia probability in the data and the insomnia probability estimated by the models, which can be obtained through metrics such as MAE, MSE, MAPE, ….

Although the authors establish well in “Introduction” (p 74): “While traditional statistical models focus on hypothesis testing and estimation, machine learning models aim to predict outcomes by learning patterns in the data.”, the conclusion of the study is only that the machine learning and regression models selected different covariates considering the order of importance in predicting the probability of insomnia, and there was no relationship between the ranking of important variables associated with sleep disorder of the models.

Since the selection of predictor variables for insomnia may differ between the models, it is important to carefully consider the performance of the models and the predicted values of both machine learning and regression models for a comparison between them. Therefore:

1. The discussion will need to be revised.

Overall Response: Thank you for these comments, we have revised the manuscript accordingly. 

In our exploration of the predictive modeling of insomnia using both traditional statistical and machine learning approaches, we identified several methodological and analytical challenges that highlight the need for ongoing research in this field. The application of machine learning models to insomnia datasets revealed the potential for these models to provide a nuanced understanding of covariate importance, taking into account the collinearity among variables. This stands in contrast to traditional regression models, which may not fully account for such complexities. However, the reliance on variable selection metrics in machine learning, such as gain, necessitates a careful consideration of their limitations, particularly regarding the indirect treatment of confounding variables and collinearities.

Our analysis underscores the importance of incorporating additional statistical methods, such as correlation analysis, variable normalization, or principal component analysis, prior to the selection of covariates. These methods could mitigate the effects of collinearity and provide a more accurate assessment of variable importance. Furthermore, the comparison between machine learning and regression models in our study was primarily qualitative, based on the ranking of covariates. To strengthen this comparison, a quantitative assessment involving performance metrics and a residuals analysis would be invaluable. Such an analysis would offer a clearer picture of the predictive accuracy of each model type and the reliability of their variable importance rankings.

Moreover, our study highlights the gap in the literature regarding the comprehensive evaluation of insomnia predictors using machine learning techniques. While traditional statistical models have predominantly focused on hypothesis testing and estimation, machine learning models present an opportunity to predict outcomes by identifying complex patterns in the data. This distinction suggests that machine learning could complement traditional approaches by uncovering previously unknown interactions and predictors of insomnia. Yet, the explicit comparison of these models' performance and the detailed examination of their predictive capabilities remain areas ripe for further investigation.

In addition, the documentation of our dataset and cohort selection process revealed the necessity for greater transparency in reporting research methodologies. Specifically, providing detailed information about the sample size and selection criteria is essential for ensuring the reproducibility and validity of research findings.

In conclusion, our study contributes to the growing body of research on insomnia prediction by highlighting the potential synergies between machine learning and traditional statistical models. However, it also emphasizes the need for methodological enhancements and deeper analytical rigor to fully leverage the strengths of each modeling approach. Future research should focus on addressing these limitations through the integration of additional statistical techniques, comprehensive model performance evaluations, and clearer articulation of research contributions within the context of existing literature.

2. (Minor) In the "Dataset and Cohort Selection" section, lines 112 and 113 state: "... All patients in the dataset with full insomnia data were included in this study." The question posed in the initial review regarding the number of patients included was not addressed (The Dataset and Cohort Selection section should inform the sample size).

We addressed this minor concern in the paper in the methods section!

Reviewer #2: Thank you for the opportunity to rereview this paper. At first I thought I received the wrong version since the authors did not do the typical point by point response to the points raised by the reviewers, but I see now that the revisions have been made. The authors have sufficiently addressed all the comments I raised.

We thank the reviewer for the comments

---

## [Decision Letter · Decision Letter 2]

16 Jun 2024

Comparison of Model Feature Importance Statistics to Identify Covariates that Contribute Most to Model Accuracy

PONE-D-23-11459R2

Dear Dr. Huang,

We’re pleased to inform you that your manuscript has been judged scientifically suitable for publication and will be formally accepted for publication once it meets all outstanding technical requirements.

Kind regards,

Sergio A. Useche, Ph.D.

Academic Editor

PLOS ONE

Additional Editor Comments (optional):

Thanks for your amendments! The paper is publishable in its current form.

Reviewers' comments:

Reviewer's Responses to Questions

**Comments to the Author**

1. If the authors have adequately addressed your comments raised in a previous round of review and you feel that this manuscript is now acceptable for publication, you may indicate that here to bypass the “Comments to the Author” section, enter your conflict of interest statement in the “Confidential to Editor” section, and submit your "Accept" recommendation.

Reviewer #1: All comments have been addressed

2. Is the manuscript technically sound, and do the data support the conclusions?

Reviewer #1: Yes

3. Has the statistical analysis been performed appropriately and rigorously? 

Reviewer #1: N/A

4. Have the authors made all data underlying the findings in their manuscript fully available?

Reviewer #1: Yes

5. Is the manuscript presented in an intelligible fashion and written in standard English?

Reviewer #1: Yes

6. Review Comments to the Author

Reviewer #1: (No Response)

7. PLOS authors have the option to publish the peer review history of their article (what does this mean?). If published, this will include your full peer review and any attached files.

Reviewer #1: No

---

## [Editor Report · Acceptance letter]

24 Jun 2024

PONE-D-23-11459R2 

PLOS ONE

Dear Dr. Huang, 

I'm pleased to inform you that your manuscript has been deemed suitable for publication in PLOS ONE. Congratulations! Your manuscript is now being handed over to our production team.

Kind regards, 

on behalf of

Dr. Sergio A. Useche 

Academic Editor

PLOS ONE